# The Role of Inflammation in The Cellular and Molecular Mechanisms of Cardiopulmonary Complications of Sickle Cell Disease

**DOI:** 10.3390/biom13020381

**Published:** 2023-02-17

**Authors:** Oluwabukola T. Gbotosho, Jahnavi Gollamudi, Hyacinth I. Hyacinth

**Affiliations:** 1Department of Neurology and Rehabilitation Medicine, University of Cincinnati, Cincinnati, OH 45267-0525, USA; 2Division of Hematology & Oncology, Department of Internal Medicine, 3125 Eden Avenue, ML 0562, Cincinnati, OH 45219-0562, USA

**Keywords:** sickle cell disease, cardiopulmonary complications, inflammation, acute chest syndrome, cardiac hypertrophy, cardiac fibrosis, diastolic dysfunction, pulmonary hypertension

## Abstract

Cardiopulmonary complications remain the major cause of mortality despite newer therapies and improvements in the lifespan of patients with sickle cell disease (SCD). Inflammation has been identified as a major risk modifier in the pathogenesis of SCD-associated cardiopulmonary complications in recent mechanistic and observational studies. In this review, we discuss recent cellular and molecular mechanisms of cardiopulmonary complications in SCD and summarize the most recent evidence from clinical and laboratory studies. We emphasize the role of inflammation in the onset and progression of these complications to better understand the underlying pathobiological processes. We also discuss future basic and translational research in addressing questions about the complex role of inflammation in the development of SCD cardiopulmonary complications, which may lead to promising therapies and reduce morbidity and mortality in this vulnerable population.

## 1. Introduction

Sickle cell disease (SCD) is the most common monogenic blood disorder, affecting approximately 100,000 Americans and millions more worldwide [1,2]. Cardiopulmonary complications are a major cause of morbidity and mortality in SCD, accounting for 32–70% of deaths [3,4,5]

Several pathophysiological processes, including anemia, hemolysis, endothelial dysfunction, and ventricular remodeling, may contribute to cardiopulmonary complications in SCD [3,4,5]. Although the etiology of cardiopulmonary complications in SCD is somewhat different from that in the general population, there are similarities in the cellular and molecular mechanisms that underlie the pathogenesis in both scenarios and that are beginning to gain prominence. Accumulating evidence has long identified chronic low-grade inflammation as a risk factor for the progression of myocardial infarction, ventricular hypertrophy, cardiac fibrosis, diastolic dysfunction, and pulmonary hypertension in the general population [6,7,8,9,10]. Recent mechanistic and observational studies on cardiopulmonary complications of SCD implicate inflammation as a major player in the onset and progression of cardiopulmonary complications in SCD [11,12,13,14,15]. These include several studies in animals and humans on the development of acute chest syndrome (ACS), cardiac hypertrophy, pulmonary hypertension (PH), cardiac fibrosis, and diastolic dysfunction [11,12,13,14]. These studies consistently suggested inflammatory pathways as a vital unifying mechanism that accompanies the structural and functional changes that occur at the onset and progression of these complications. Thus, integrating therapies that balance the pro-inflammatory and anti-inflammatory processes contributing to the chronic inflammatory state in SCD may provide opportunities for novel therapies that could be easily incorporated into the existing treatment options available to SCD patients. The recent advances in cellular and molecular mechanisms of cardiopulmonary complications of SCD, along with the complex interplay between inflammation and the unique cardiac pathology of SCD such as acute chest syndrome, pulmonary hypertension, diastolic dysfunction, and cardiac hypertrophy, are described in this review (Figure 1).

## 2. Inflammation and Acute Chest Syndrome

Acute chest syndrome (ACS) is a pulmonary complication of SCD and the second leading cause of mortality and morbidity in both adults and children with SCD [16]. It is defined as the presence of fever and/or new respiratory symptoms such as cough, chest pain, and presence of a new pulmonary infiltrate on chest X-ray [16]. Risk factors for ACS include younger age, severe SCD genotypes (SS or Sβ^0^ thalassemia), lower fetal hemoglobin concentrations, inflammation, higher steady-state white blood cell counts, history of asthma, and tobacco-smoke exposure [17,18]. The major causes known to trigger ACS include respiratory infection, pulmonary infarction, or fat embolism; however, no specific cause can be found in up to 30% of cases [17]. At a cellular level, an inciting trigger such as an infection permits increased adhesion of leukocytes (neutrophils) to the lung microvasculature, generation of cytokines, coupled with interactions with other cellular components such as platelets. This results in local hypoxemia and changes in rheology of the red blood cells (RBCs). This further facilitates interactions between RBCs, vascular endothelium, and leukocytes, resulting in increased oxidative stress, vaso-occlusion, and tissue hypoxia. These events in turn result in additional recruitment of leukocytes and other cellular components to the site, thereby amplifying the inflammatory cascade, resulting in a “vicious” cycle of lung injury and hypoxemia [19,20].

Evidence for heightened inflammation in the pulmonary microenvironment during ACS comes from human studies which show that children with ACS have high levels of IL-6, IL-8, CCL2, and CCL3 in their sputum [12]. These cytokines, particularly CCL2 and CCL3, have been shown to recruit leukocytes, particularly neutrophils, via upregulation of platelet activating factor (PAF) and leukotriene B-4 (LTB4). The neutrophils firmly adhere to the endothelium and become activated as assessed by shedding of CD62L and upregulation of CD11b [21]. Upregulation of CD11b in arrested leukocytes enables their interaction with GPIbα expressed on platelets [22]. Arrested neutrophils can also interact with platelets via PSGL-1 on neutrophils binding to P-selectin on platelets. This is evidenced by autopsy studies which show the presence of large neutrophil–platelet aggregates and platelet-laden aggregates in pulmonary vasculature in patients with ACS [19,23]. Indeed, preclinical studies that inhibit P-selectin and GPIbα interactions show fewer leukocyte–platelet aggregates [24], highlighting the importance of neutrophil and platelet heterotypic interactions in pathogenesis of ACS. Furthermore, a study by Ghosh et al., in a Townes sickle cell mouse model, showed that P-selectin in both platelet and endothelium compartments played a dominant role in promoting heme-induced ACS in SCD [25].

Hemolysis is a pathological feature of SCD that releases free hemoglobin and heme into the circulation due to RBC sickling and lysis, leading to the activation of inflammatory signaling pathways and vascular inflammation [26,27,28,29]. The release of free heme and cell-free hemoglobin also results in activation of neutrophils and generation of neutrophil extracellular traps (NETs) [30], iron-based generation of reactive oxidative species (ROS) with subsequent oxidization of membrane lipids [31], depletion of nitric oxide [32], and endothelial cytoskeleton remodeling resulting in barrier dysfunction [33]. Furthermore, plasma free heme and other markers of hemolysis have been associated with increased odds of developing ACS in children with SCD [34]. Additionally, a mutation in the heme-oxygenase 1 (HMOX1) short (GT)n repeat promoter that confers stronger inducibility of HMOX-1, the rate-limiting enzyme that degrades heme, was associated with a reduction in the rate of hospitalization for ACS in children with SCD [35]. These studies were validated in both Townes and Berkeley SCD mouse models using extracellular heme as a trigger for ACS. Heme exposure causes respiratory failure due to rapid hypoxemia and death, mimicking some of the events associated with ACS in SCD patients [36]. Treatment of SCD mice with D3T (3H-1,2-dithiole-3-thione), an activator of nuclear-factor erythroid 2 like 2 (NRF2), which controls HMOX1 expression, reduced lethality in a model of heme-induced ACS in SS mice [37]. Additionally, treatment with hemopexin, the plasma heme-binding protein, abrogates lung injury and mortality in a chlorine (Cl_2_)-inhalation model of inducing ACS [38]. These studies suggest that therapies that target the product (heme) or molecular consequence(s) of hemolytic pathways may offer protection from ACS in SCD.

## 3. Inflammation and Pulmonary Hypertension

Pulmonary hypertension (PH) is an independent risk factor for early death in SCD patients [39]. Its estimated prevalence, as assessed by right heart catheterization (RHC), is about 6–10% [40], although this estimate relied on an older definition used to diagnose PH. Per the most recent guidelines, PH is now defined as mean pulmonary artery pressure of >20 mm Hg in conjunction with pulmonary artery wedge pressure of ≤15 mm Hg and a pulmonary vascular resistance (PVR) of ≥3 Wood units (WU). A diagnosis of isolated post-capillary PH is made when PVR is <3 WU, whereas a PVR of ≥3 WU is supportive of pre-capillary PH [41]. In SCD, pre-capillary, post-capillary, and pulmonary thromboembolic PH or a combination can exist. Risk factors for PH include chronic intravascular hemolysis, pulmonary thrombosis or embolism, and heart failure [42,43]. 

The development of PH in SCD is complex and involves pulmonary vascular endothelial dysfunction, smooth muscle cell (SMC) proliferation and resistance to nitric oxide (NO) adventitial fibroblast accumulation, and inflammation. Interestingly, one of the unique features of PH in SCD is the presence of iron in pulmonary macrophages, a feature that is not seen in other forms of PH. An autopsy study of lung samples from SCD patients with PH and RV failure found peripheral monocytes and macrophages accumulating in the perivascular and alveolar regions of the lungs [44]. These macrophages had extensive iron accumulation concomitantly with the expression of HMOX1, ET-1, and IL-6 [44]. This suggests that, in pathological diseases with hemolysis such as SCD, circulating immune cells may be recruited into the lungs for heme degradation. However, this immune response may become maladaptive over time, as accumulated iron may contribute to oxidative stress, alter the redox balance, or induce transdifferentiation of resident lung macrophages and other alveolar cells. This underscores an important role for intravascular hemolysis in the pathogenesis of PH [45,46,47,48]. SCD is characterized by increased stress erythropoiesis as a compensatory mechanism for anemia, which increases the number of reticulocytes and younger RBCs in circulation. During hemolysis, these young RBCs release a large amount of arginase into the plasma [49]. This plasma arginase consumes plasma L-arginine, the substrate required for NO production by endothelial cells, and, in conjunction with the consumption of endothelial NO by cell-free plasma Hb, reduces NO bioavailability [50]. The depletion of NO affects intracellular calcium signaling that leads to dephosphorylation of myosin protein, preventing smooth muscle relaxation [51]. The depletion of NO also results in leukocyte recruitment via increased surface expression of leukocyte adhesion proteins such as E-selectin, VCAM, and ICAM-1 [52,53] and results in smooth muscle proliferation and vascular remodeling [54]. Heme-related generation of ROS decreases availability and or activation of soluble guanylyl cyclase or its regulators such as cytochrome b5 reductase 3 (CYB5R3), which can result in poor vasodilation of pulmonary vasculature, increasing the risk of pulmonary hypertension [55,56]. 

Cell-free plasma hemoglobin and heme can also independently activate platelets and neutrophils via a TLR4 signaling mechanism [30], resulting in further inflammation. In addition, cell-free hemoglobin also generates ROS, which furthers endothelial dysfunction and activates the coagulation system [57]. Chronic hemolysis also promotes transition of pulmonary endothelial cells to a mesenchymal or smooth muscle cell type and contributes to vascular remodeling [58]. Thus, heme exposure results in pathological endothelial activation, increased recruitment of leukocytes and depletion of protective mechanisms that preserve vascular integrity. 

Mechanistically, endothelial dysfunction results in increased production of vasoconstrictors such as endothelin-1 (ET-1). Heme-related endothelial dysfunction can deplete peroxisome proliferator-activated receptor γ (PPARγ), which plays an active role in suppressing ET-1 production by regulating the level of microRNAs (miRs) such as miR-98 [53]. Lower levels of miR-98 are associated with increased ET-1 production and endothelial dysfunction [52]. Exposure to heme also results in increased production of placenta growth factor (PIGF) by erythroid cells via the erythroid Krüppel-like factor (EKLF) [59] and NRF2-antioxidant response signaling [60]. PIGF is an angiogenic factor that activates endothelial cells to secrete ET-1 [61]. In an elegant study, overexpression of erythroid-specific PIGF in normal mice up to the levels seen in sickle cell mice resulted in an increase in the production of ET-1, which correlated with increased right ventricular pressure and pulmonary arteriolar thickening [62]. Elevated ET-1 and PlGF levels also correlate with severity of PH in patients with SCD [62]. PIGF was shown to activate expression of hypoxia-inducible factor 1α (HIF-1α), independently of hypoxia, which in turn can stimulate expression of ET-1, which is involved with the development and severity of PH in SCD [61]. 

Indeed, these cellular and molecular mechanisms have informed the current therapeutics usually used in patients with pulmonary hypertension such as endothelin receptor (ETR) antagonists (Bosentan, Ambrisetan), those which prevent the degradation of cyclic guanosine monophosphate (cGMP) (Riociguat and Sildenafil), vasodilators (Epoprostenol), and anticoagulant (warfarin), among others. Clinical trials using hemopexin, a scavenger molecule that removes heme from circulation, are underway in humans and have shown promising results in murine models [63]. Unfortunately, trials with ETR antagonists [64], Riociguat [65], and Sildenafil [66,67] were either limited by small sample size or adverse side effects, underscoring the need to better understand the pathology and the need for larger clinical trials researching PH in SCD. 

## 4. Inflammation and Pulmonary Thrombosis Embolism 

Accumulating evidence from human studies discussed below suggests that inflammation is a risk factor for thrombosis. It is therefore not surprising that a retrospective study found that the prevalence rate of venous thromboembolism (VTE) in adults with SCD was 25% and was associated with increased rates of recurrence and mortality [68,69]. Interestingly, the risk of pulmonary embolism (PE) is higher than the risk of deep vein thrombosis (DVT) [69,70,71] suggesting that thrombosis may occur more ‘in situ’ in pulmonary vasculature of individuals with SCD. Risk factors for VTE include elevated leukocyte count [72], severe phenotype as defined by >3 hospitalizations annually for vaso-occlusive crisis, presence of SCD variant genotypes, elevated tricuspid regurgitation jet velocity (TRJV) ≥2.5 m/s [68], elevated body mass index, and prior splenectomy [68,73,74]. Even in those with lower hospitalizations, the cumulative incidence rate of VTE was at 6.8% compared to 1.6% in individuals who had similar number of hospitalizations for asthma exacerbation [69], suggesting that intrinsic pathology, in addition to external risk factors, plays a role in development of PE and/or DVT. 

Several cellular and molecular pathways are perturbed in SCD that leads to a prothrombotic state. Chronic hemolysis results in the release of intracellular molecules known as danger (or damage)-associated molecular patterns (DAMPs) [75]. For example, one of the most studied DAMPs or alarmins, high-mobility group box 1 (HMGB1), is significantly elevated in the plasma of SCD patients and mice at baseline compared to controls [76,77]. VOC episodes further increased HMGB1 levels in SCD patients, or acute sickling induced following hypoxia-reoxygenation in mice [76]. Furthermore, treating the TLR4 reporter cell line with plasma from SCD patients increased TLR4 receptor activity, suggesting that HMGB1 contributes to TLR4 signaling in SCD [76]. Elevated circulating HMGB1 is associated with the platelet nucleotide-binding domain, leucine-rich-containing family, and pyrin-domain-containing-3 (NLRP3) activation, which is mediated through the TLR4 and Bruton tyrosine kinase signaling pathways [77,78]. Another study in murine macrophages has shown that cell-free hemoglobin and free heme act in synergy with HMGB1 to activate proinflammatory cytokine production in wild-type murine macrophages, and treatment with hemopexin abolishes this interaction [79]. Furthermore, treatment with hemopexin significantly suppressed the synergistic production of proinflammatory cytokines, suggesting an anti-inflammatory property of hemopexin [79]. This anti-inflammatory ability of hemopexin, in addition to its heme-scavenging function, may provide another potential therapeutic option for addressing inflammation in SCD. DAMPs have also been implicated in endothelial dysfunction [80], activation and recruitment of leukocytes [30,75] and inflammation [75], which shift the balance to a more prothrombotic state in SCD. The characteristic changes in RBC rheology contribute to the formation of venous thrombi that have a denser fibrin network and a friable thrombus [81]. In addition, red-cell-derived microparticles contribute to thrombin generation via activation of Factor XI. Indeed, red-cell-derived microparticles are associated with increased markers of coagulation activation [82]. Activated platelets promote inflammasome activation and generation of EVs, which can lead to formation of heterotypic aggregates and occlusion of the lung’s microvasculature [83]. DAMPs can also activate neutrophils and monocytes, which can result in increased TF expression [84], NET formation, endothelial dysfunction, and more inflammation [30], which have been linked to thrombus generation and propagation in non-SCD models [85]. Endothelial dysfunction from heme exposure results in the upregulation of adhesion molecules that attract neutrophils and platelets [30,33,80], and increased expression of TF and VWF, which can contribute to pulmonary thrombosis [86]. 

The exact molecular mechanisms resulting in thrombus formation in lungs in SCD are not well studied and may involve mechanisms that either increase procoagulant proteins (such as TF, VWF, thrombin) [86,87], decrease anticoagulant proteins (like low protein C and S) [88,89], and/or decrease fibrinolysis [81]. There is some evidence that abrogation of TF using anti-TF antibody reduces thrombus formation in a sickle cell mouse model, suggesting an important contribution of TF to thrombus generation in SCD. In addition, genetic or immunologic interventions that modulated expression of protein C and thrombin also diminished thrombus formation [90]. Data supporting the role of contact pathways leading to thrombosis in SCD are very sparse; however, potential plausible sources include neutrophil nuclear content, specifically nuclear DNA and histones which can initiate coagulation by activating Factor XII but also amplify thrombin-dependent factor XI activation [85,91]. Partial support for this comes from observation that inducing neutropenia results in decreased thrombosis burden in an arterial thrombosis model [90]. Thus, several cellular and molecular mechanisms may be at play in the pathogenesis of thrombosis. 

## 5. Inflammation and Reactive Airway Disease or Airway Hyper-Activity (AHR)

Reactive airway disease or AHR is common among adults and children with SCD [92,93] and can occur independent of clinical asthma. Studies show that up to 77% of children demonstrate a positive methacholine challenge test (MCT) [92,94]. Interestingly, the severity of AHR correlates with high LDH, suggestive of a critical role played by hemolysis and disease severity [95]. Indeed, one study did show that AHR was more common in those with the HbSS genotype and was predictive of increased risk of ACS and vaso-occlusive crisis [96]. From a pathophysiology perspective, AHR is characterized by bronchial hypersensitivity to stimuli, airway and lung inflammation, abnormal leukocyte recruitment, and airway and vessel wall thickening [97,98,99]. Chronic hemolysis and its byproducts may drive systemic inflammation and result in increased lung/airway inflammation. Indeed, in one study with SCD mice, even prior to sensitization with an allergen, there was evidence of increased airway inflammation, increased lymphocytes in bronchoalveolar fluid (BAL), granulocyte-colony stimulating factor, interleukin 5 (IL-5), IL-7, and chemokine (C-X-C motif) ligand (CXCL)1, and lung T cell infiltration [100]. Mice exposed to specific allergen recapitulated specific features of an asthma-like phenotype, including increased immunoglobulin E (IgE), IL-6, and IL-13 in serum and increased bronchial hyperresponsiveness to methacholine [100]. Another study corroborated the findings of increased IgE and airway inflammation, as evidenced by eosinophil infiltration, vessel wall thickening, and increased concentrations of transforming growth factor beta (TGF-B) [101]. SCD mice with PIGF deficiency showed decreased airway inflammation, leukotriene, and IL-13-mediated immune responses, suggesting an important role of PlGF signaling pathways in AHR [96]. Thus, multiple pathways are at work that make individuals with SCD susceptible to allergens and environment pollutants.

## 6. Inflammatory Mediators and Cardiac Hypertrophy

Both concentric and eccentric hypertrophy have been reported in children and adults with SCD [102]. These changes in cardiac structure and function start in early childhood and worsen with age [103,104].The enlargement of the heart begins as a compensatory myocardial remodeling in response to anemia [105]. The cardiomyocytes and capillary networks in the heart become enlarged to increase oxygen supply, leading to increased cardiac output at rest [105]. This elevated cardiac output also becomes exaggerated during exercise due to an increase in cardiac stroke volume in response to the increased oxygen consumption, indicating altered hemodynamics in SCD patients [105,106]. Restrictive cardiomyopathy can also co-exist with anemia-induced elevated cardiac output [103,104]. Morphological abnormalities of sickle RBCs, such as polymerization and aberrant membrane transport properties, auto-oxidative ROS generation, and ischemia-reperfusion injury, may also contribute to cardiac remodeling [107,108]. 

Studies have suggested that endothelial dysfunction and increased plasma markers of inflammation contribute to cardiac hypertrophy in SCD mouse models [109,110]. This may be due to the pre-activation of immune cells, including monocytes and endothelial cells, in SCD [111,112,113,114]. There is also an increase in the mRNA and protein expression of pro-inflammatory markers such as TNF-α, IL-1, IL-6, MIP-1b, and soluble endothelial adhesion molecules [113,115,116]. The heart, like the other organs, is exposed to inflammatory insults from these circulating proinflammatory cytokines. Given that elevated systemic inflammation is associated with cardiac abnormalities in the general population [6], it is possible that the observed excessive systemic inflammation in individuals with SCD could contribute to cardiac pathology in SCD. 

In addition to pro-inflammatory cytokines, products of hemolysis such as heme and other DAMPs may further perturb the homeostatic state in the heart, thereby perpetuating the vicious cycle of inflammation and cardiac pathobiology described in the last paragraph. For instance, heme released into the circulation during hemolysis triggers several inflammatory pathways in SCD that contribute to organ damage [29,36,80]. In fact, a comparison of organ-specific expression patterns of HMOX1 in SCD mice treated with heme revealed that the heart has one of the highest expressions of HMOX1 [117]. This suggests that cardiac cells can uptake circulating heme and metabolize it. The potential problem with this process is that excess iron produced from heme breakdown and deposited in the heart can activate oxidative and apoptotic pathways. A recent study showed that heme-induced upregulation of HMOX1 promotes cardiac ferroptosis in SCD mice as well as the expression of cardiac hypertrophy genes [13], although T2* cardiac magnetic resonance imaging measurement of cardiac iron showed that iron overload is rare even in chronically transfused SCD patients [118,119]. Another study showed that increasing circulating heme significantly elevated plasma IL-6 and the expression of cardiac hypertrophy markers in Townes sickle cell mice [15]. These studies underscore the importance of hemolysis in the pathogenesis of cardiac hypertrophy. Another way that inflammation may contribute to cardiac pathology is through a complex interaction with the coagulation system. Increasing evidence in SCD has shown a link between vascular inflammation and hypercoagulation via activated intrinsic and extrinsic coagulation pathways, which may contribute to organ pathology [120,121]. A study by Sparkenbaugh et al. showed that short-term pharmacological inhibition of FXa in Berkeley sickle cell mice attenuated plasma IL-6 and cardiac hypertrophy [109]. Similarly, genetic inhibition of circulating FII in SCD mice improved right ventricle hypertrophy and dilatation, suggesting that increased thrombin generation or activity in SCD is a significant contributor to cardiac pathophysiology [110]. These studies suggest that perturbation in the coagulation system and the attendant vascular damage in SCD may contribute to inflammation, a key component of various mechanisms involved in cardiac dysfunction. 

The inflammatory signaling pathway may also modify cardiac remodeling in SCD via the complex biological role played by PlGF. PlGF is an angiogenic cytokine that plays a role in the survival of endothelial cells and monocytes and in cardiovascular health [114,122]. PlGF is crucial in the early inflammatory response needed for adaptive hypertrophic cardiac remodeling due to pressure overload [123,124]. Although studies have shown both beneficial and deleterious roles of PlGF in the heart [125,126,127], its expression is elevated in the plasma of SCD patients and linked with disease severity [62,113]. Furthermore, PlGF mRNA and protein expression were found to be elevated in the hearts of sickle cell mice at baseline and upon exposure to heme [117]. The role of PlGF in cardiopulmonary complications of SCD may be via an indirect effect on endothelial cells, fibroblasts, and monocytes in the heart, which are already primed for an exaggerated response due to a pro-inflammatory microenvironment mediated by cytokines such as IL-6. Nevertheless, no mechanistic studies have clearly delineated this intriguing role of PlGF in cardiac pathology in SCD by examining these cardiac cells individually.

## 7. Inflammation and Diastolic Dysfunction

Diastolic dysfunction occurs in both children and adults with SCD. It has been associated with anemia, older age, higher creatinine levels, exercise impairment, increased LV mass, low sleep or waking oxygen saturation, and diffuse myocardial fibrosis [11,102,128,129,130]. Though the causative sequence is not well defined, diastolic dysfunction is also an independent risk factor for death in SCD patients [103,131]. Increased doppler echocardiography ratio of mitral valve inflow (E) velocities (E) over peak early diastolic annular velocity (E′) >8.2 has been shown to predict diastolic dysfunction in SCD patients [11,128,132]. Furthermore, diastolic dysfunction has been linked to the overexpression of interleukin-18 (IL-18), -L-fucosidase A2 (FUCA2), and thyroid hormone transporter (SLC16A2) in SCD patients’ peripheral blood mononuclear cells (PBMCs) [11]. This finding was validated in mouse models, with results showing elevated expression of these genes in the myocardium of sickle cell mice compared to controls [11]. Although diastolic dysfunction in SCD involves multiple complex pathophysiological mechanisms, this report on IL-18 suggests the involvement of inflammation, which is chronic in SCD, either as a primary mechanism or as a secondary mechanism to cardiac remodeling due to the hyperdynamic state.

## 8. Inflammation and Cardiac Arrhythmia

Cardiac arrhythmia (CA) is defined as a disruption in the normal activation of the heart or an irregular heartbeat rhythm that is either too slow (60 beats per minute) or too fast (>100 beats per minute) [133]. Some forms of CA, including sinus arrhythmia, are considered to be benign; however, the presence of structural heart defects such as LV dysfunction or genetic arrhythmia syndromes, including long or short QT syndrome, increases the severity of CA and could result in heart failure or sudden cardiac death [133,134]. Cardiac arrythmias have been reported as the cause of death in 7.4% of in-hospital deaths in adult African Americans with sickle cell trait [135] and in 14% of SCD patients. Cardiac arrhythmias have also been implicated in some sudden deaths recorded in SCD patients with a prolonged corrected QT interval (QTc), which is independently associated with an increased risk of death [106,136,137]. In a recent study, elevated expression of IL-18 in PBMCs of SCD patients was associated with longer QTc intervals and increased mortality risk [138]. Consequently, chronic inhibition of IL-18-binding protein in a sickle cell mouse model attenuated IL-18-mediated ventricular tachycardia and improved diastolic function, suggesting a link between cardiac inflammation and arrhythmias in SCD [138].

## 9. Inflammation and Cardiac Fibrosis

Autopsy studies and studies in living SCD patients have shown the presence of both diffuse and transmural fibrosis [102,139,140]. The cellular and molecular mechanisms underlying cardiac fibrosis in SCD are not completely understood. However, gene-expression profiles of heart tissue isolated from Berkeley sickle cell mice revealed elevated expression of genes involved in oxidative stress, angiogenesis, and TGF-β signaling, which correlated with imaging and histology data demonstrating diffuse cardiac fibrosis and diastolic dysfunction [141]. Another study in SCD mice showed that sustained neutralization of the IL-18-binding protein ameliorated cardiac fibrosis [138] (Figure 2). In a small observational study of SCD patients, early initiation of disease-modifying therapy such as hydroxyurea and chronic transfusion was shown to prevent diffuse myocardial fibrosis and diastolic dysfunction [142]. Although the underlying specific mechanism(s) of how these therapies ameliorate the development of cardiac fibrosis remains unknown, there is a need for detailed mechanistic studies that elaborate on the link between inflammation and other pathways that may be involved in the development of cardiac fibrosis in SCD. 

## 10. Conclusions and Future Perspectives

Inflammation involves complex cellular and molecular pathways that have both beneficial and harmful effects, and it plays a major role in the pathogenesis of cardiopulmonary complications in SCD. Inflammation is also a common denominator in the development of ACS, cardiac fibrosis, PH, and diastolic dysfunction, as reviewed here (Table 1). Several studies have shown that chronic systemic inflammation as present in SCD contributes to a cascade of events that results in tissue injury, the generation of ROS, and endothelial dysfunction [143,144,145]. Additionally, inflammatory cells such as monocytes are in a state of pre-activation due to elevated levels of circulating inflammatory cytokines, which contribute to the chronic inflammation in SCD by amplifying the production and signaling of these cytokines [62,113,144]. Therefore, addressing cardiopulmonary manifestations of the disease by modulating inflammatory pathways using targeted therapies could offer a novel approach to reducing the risk of developing these complications. This would involve well-designed and detailed experiments that include investigating the role of genetic polymorphisms in the regulation of blood levels of circulating cytokines, inflammatory markers, and their signaling pathways in SCD. Additional investigation will also examine how genes encoding the production of inflammatory mediators are regulated and potentially silenced. Furthermore, a detailed examination of how the inflammatory SCD microenvironment modifies the functions of immune cells as well as the response of these immune cells is needed. Identification of plasma inflammatory biomarkers with prognostic value for determining SCD patients at risk of cardiopulmonary complications may also be an additional direction of investigation that could be beneficial. The adoption and adaptation of advanced non-invasive imaging technology that provides both structural and functional information about the cardiopulmonary landscape in SCD patients may provide a novel approach to documenting cardiac remodeling in SCD patients. For example, hybrid positron emission tomography (PET) and magnetic resonance imaging (MRI) using the uptake of the radiotracer 18F-fluorodeoxyglucose (18F-FDG) have been used to identify and characterize vascular inflammation and different stages of atherosclerosis that are not yet detectable by other forms of structural imaging such as cardiac computed tomography and echocardiography [146,147]. This technique may be useful and could be adapted to SCD, where both inflammation and vascular dysfunction are present and linked with the development of cardiac hypertrophy, PH, and diastolic dysfunction. Because inflammation may precede the development of cardiopulmonary complications in SCD patients, extending the current guidelines for screening, diagnosis, and management of cardiopulmonary complications in SCD patients [41], by adding biomarkers of inflammation linked with cardiopulmonary complications, is needed. This may improve risk stratification and help with better identification of patients needing more urgent therapy, thus preventing the progression to irreversible organ damage.

Overall, early diagnosis of cardiopulmonary complication through early recognition and application of molecular risk factors before irreversible organ damage occurs would contribute to a better quality of life, as SCD patients are now living longer due to the availability of disease-modifying therapies.

## Figures and Tables

**Figure 1 biomolecules-13-00381-f001:**
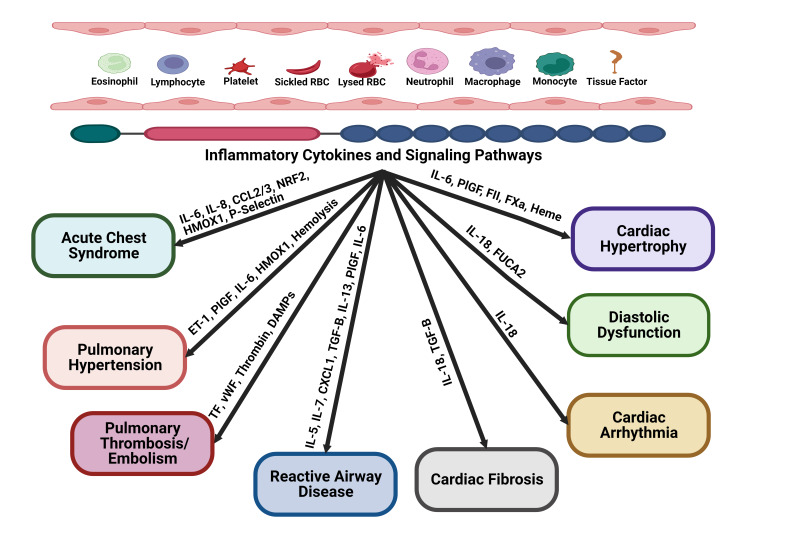
A visual representation of the inflammatory mediators that may be involved in the cardiopulmonary complications of sickle cell disease. Interleukin 6 (IL-6), IL-5, IL-7, IL-18, IL-13, Tissue factor (TF), C-C motif chemokine ligand 2 (CCL2/3), reactive oxygen species (ROS), nuclear-factor erythroid 2 like 2 (NRF2), heme-oxygenase 1 (HMOX1), endothelin-1 (ET-1), placental growth factor (PlGF), von Willebrand factor (VWF), danger-associated molecular patterns (DAMPs), chemokine (C-X-C motif) ligand 1 (CXCL1), Factor II (FII), Factor Xa (Fxa), and transforming growth factor (TGF-β).

**Figure 2 biomolecules-13-00381-f002:**
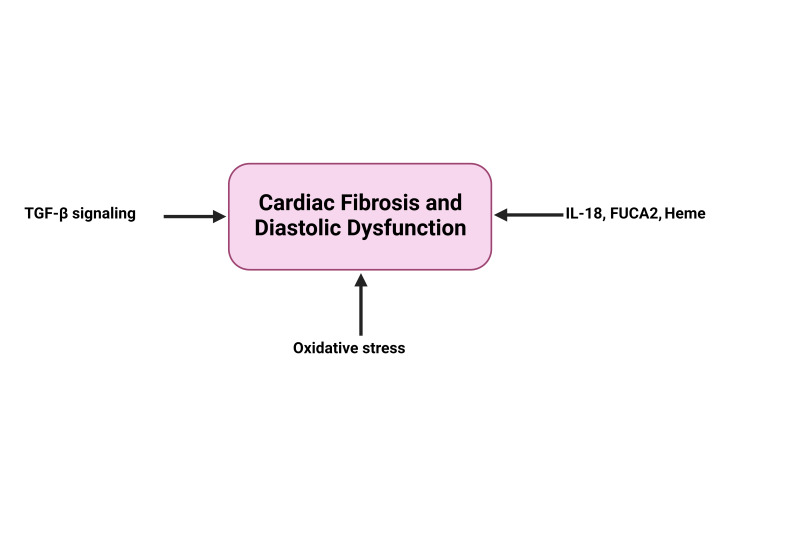
Summary of the inflammatory mechanisms associated with cardiac fibrosis and diastolic dysfunction in animals with SCD (with the relevant reports that describe them) addressed in the current review manuscript. TGF-β: transforming growth factor beta, IL-18: interleukin-18, and FUCA2: -L-fucosidase A2.

**Table 1 biomolecules-13-00381-t001:** Summary of inflammatory mediators and potential novel treatments in cardiopulmonary complications in sickle cell disease.

Disease Complication	Major Contributors	Potential Novel Treatments That May Target Major Inflammatory/Anti-Inflammatory Pathways	Citations
1. Acute Chest Syndrome (ACS)	Free heme, heme oxygenase (HMOX-1), neutrophil and platelet interactions, p-selectin	Glyco-protein Ibalpha inhibitor (CCP-224) [24] D3T (3H-1,2-dithiole-3-thione) [37] Hemopexin [38]	Anea [24], Jimenez [25], Ghosh [26], Bean [36], Ghosh [37], Ghosh [38], Alishlash [39]
2. Pulmonary hypertension	Endothelial dysfunction, hemolysis, decreased NO, increased placenta growth factor (PIGF), PPAR alpha and PPAR gamma	Hemopexin [38] BAY 54-6544 [55]	Jang [53], Wood [55], Gonzales [58], Hsu [47], Morris [50] Perelman [109], Selvaraj [110], Potoka [56], Buehler [63]
3. Pulmonary thrombosis	NETs, DAMPs, tissue factor upregulation, lower protein S and C endothelial dysfunction	Anti-TF antibody	Sparkenbaugh [105] Whelihan [85] Faes [77] Solovey [82]
4. Cardiac hypertrophy	ROS, endothelial dysfunction, hemolysis, hypercoagulation, PIGF, IL-6, heme	Rivaroxaban [104]	Sparkenbaugh [104] Bakeer [137] Gbotosho [15] Menon [13] Arumugam [106]
5. Diastolic dysfunction and cardiac arrhythmia	IL-18, FUCA-2	Anti-IL-18-binding protein [133]	Duarte [11] Gupta [134]

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
