# Peer review of "The Role of Inflammation in The Cellular and Molecular Mechanisms of Cardiopulmonary Complications of Sickle Cell Disease"

_biomolecules, 2023, doi:10.3390/biom13020381_

Round 1

Reviewer 1 Report

In this manuscript, the authors reviewed how inflammation contributes to cardiopulmonary complications of sickle cell disease (SCD), including pulmonary hypertension, acute chest syndrome, pulmonary thromboembolic disease, reactive airway disease, cardiac arrhythmias and hypertrophy, cardiac fibrosis, and diastolic dysfunction. The manuscript is very interesting and well-written and proposes novel mechanisms that may be targeted to prevent the development of or progression of cardiopulmonary manifestations of SCD.

1)      Figure 1 and Table 1 are useful. But an additional 1-2 figures or tables would be helpful to supplement the pathophysiology being described in the text.

2)      On page 6, starting at line 285, the authors describe heme-induced upregulation of HMOX1 promoting cardiac ferroptosis in the murine model of SCD. They should describe that cardiac iron overload does not significantly contribute to cardiac complications in patients with SCD.

3)      There are some grammar errors that should be corrected.

Author Response

Reviewer 1

In this manuscript, the authors reviewed how inflammation contributes to cardiopulmonary complications of sickle cell disease (SCD), including pulmonary hypertension, acute chest syndrome, pulmonary thromboembolic disease, reactive airway disease, cardiac arrhythmias and hypertrophy, cardiac fibrosis, and diastolic dysfunction. The manuscript is very interesting and well-written and proposes novel mechanisms that may be targeted to prevent the development of or progression of cardiopulmonary manifestations of SCD.

1)      Figure 1 and Table 1 are useful. But an additional 1-2 figures or tables would be helpful to supplement the pathophysiology being described in the text.

Author’s Response: We thank the reviewer for this helpful comment. We have now added one more figure to the manuscript.

2)      On page 6, starting at line 285, the authors describe heme-induced upregulation of HMOX1 promoting cardiac ferroptosis in the murine model of SCD. They should describe that cardiac iron overload does not significantly contribute to cardiac complications in patients with SCD.

Author’s Response: We have modified this description with the added statement on cardiac iron, as highlighted in yellow below. A recent study showed that heme-induced upregulation of HMOX1 promotes cardiac ferroptosis in SCD mice as well as the expression of cardiac hypertrophy genes 14, although T2* cardiac magnetic resonance imaging measurement of cardiac iron showed that iron overload is rare even in chronically transfused SCD patients 119, 120.

Author Response

Reviewer 2

This is a thorough review of the role of inflammation and the mechanisms mediating cardiopulmonary problems in sickle cell disease. However, several mistakes and the lack of discussion of the role of HMGB1 in sickle cell disease are concerning, considering there is a wealth of non-sickle cell data showing that the impact of HMGB1 on the cardiopulmonary system can be quite profound. The mistakes and deficiencies will prompt a knowledgeable reader to wonder whether the references were gathered with care and attention to detail or just thrown together with a cursory view of the abstracts. It should be recognized that the acronym, DAMP, can stand for danger- or damageassociated molecular pattern molecules. As the field uses both, it is likely that searching for one without the other will yield incomplete information. For example, searching only for sickle cell and danger-associated molecular pattern molecules yields 5 references, all dealing with reticulocyte damps. In contrast, searching for sickle cell and “damage-associated molecular pattern” yields 8 references. These references include Vogel and Thein's work (PMID: 30333099: 29383704), with additional references investigating mechanisms associated with hemolysis. Vogel and Thein show that platelet activation increases the release of platelet HMGB1, which binds to platelet TLR4 receptors, which increases not only the risk of venous thrombosis but also the activation of the platelet inflammasome, which increases the release of IL-1B. Vogel and Thein argue that this mechanism places the platelet at the interface of thrombosis and inflammation in sickle cell, this review’s primary topic. The authors mention DAMPs in their review as a major topic yet do not list any of the classical damps known to increase vascular and cardiopulmonary inflammation. Reference 80, attached to DAMPs in the review in a picture, does not contain the acronym DAMP. There are many different DAMPs, but the closest the review comes to mentioning a specific damp is when is discusses tissue factor (TF). From a very brief review of the literature, it appears as if only a few investigators considered TF to be a damp molecule. The sickle cell research community is overly focused on hemolysis. To most investigators, cf-Hb rates of hemolysis and heme are the biggest problems facing the sickle cell patient. However, isn’t SCD characterized by ischemia-reperfusion injury and hypoxiareoxygenation injury – what are the cellular consequences of IRI or HRI – cell injury and dead and dying cells? Dead and dying cells release HMGB1. Curious, I searched for HMGB1 and heart and found 561 references. So if HMGB1 is important to heart inflammation in the general population, it should also be important to sickle cell patients who have increased plasma HMGB1 levels during baseline and crisis (1, 2). Indeed, Xu et al. (2) suggest that showing that HMGB1 accounts for as much as 70% of the plasma’s ability to activate TLR4. Emerging clinical data indicates hemopexin reduces vaso-occlusive crises, suggesting that heme plays a major role in crises. However, one of the first papers examining hemopexin in sickle cell disease discovered that cf-Hb interacts with HMGB1 to form an inflammatory complex that binds to TLR4, which induces synergistic increases in cytokine generation by macrophages (3). Importantly, cell culture studies showed that the interaction between cfHb and HMGB1 could be disrupted by hemopexin (3).

Author’s Response: We thank the reviewer for noting that HMGB1 was not discussed in the review. We have included this in the revised copy on page 5 from lines 196-205 and added additional references. As noted by the reviewer, the "D" in DAMP is interchangeably used as "danger" or "damage." We have now included "danger" in the definition. We are aware of the established link between HMGB1 and heart inflammation in the general population; however, this review is specifically focused on cardiopulmonary complications in SCD. Given the paucity of data investigating the role of HMGB1 specifically in cardiopulmonary complications of SCD, we included references to studies in the general population where necessary. We did, however, performed a new search based on the reviewer’s recommendation and found that there are no publications exploring the role of cardiopulmonary complications in SCD.

Emerging clinical data indicates hemopexin reduces vaso-occlusive crises, suggesting that heme plays a major role in crises. However, one of the first papers examining hemopexin in sickle cell disease discovered that cf-Hb interacts with HMGB1 to form an inflammatory complex that binds to TLR4, which induces synergistic increases in cytokine generation by macrophages (3). Importantly, cell culture studies showed that the interaction between cfHb and HMGB1 could be disrupted by hemopexin (3).

Author’s Response: We found that reference 3, mentioned by the reviewer above and listed at the end of the comments, is an ASH abstract published in 2007. The abstract only reported levels of plasma HMGB1 in 12 SCD patients at baseline and during a crisis. We couldn’t find the full publication of this work, and as such, we are unable to comment on the interaction between HMGB1 and cytokine production by macrophages in SCD.

Page 4 – line 139 Cell-free hemoglobin reacts with NO to form nitrates and limits the availability of arginine, …. This statement as written is incorrect. Cf-Hb does nothing directly to arginine availability. Reticulocytes contain arginase, which is released upon hemolysis. Arginase degrades arginine, not cf-Hb.

Author’s Response: We thank the reviewer for this helpful comment. We have modified the statement, which now reads as follows: "SCD is characterized by increased stress erythropoiesis as a compensatory mechanism for anemia, which increases the number of reticulocytes and younger RBCs in circulation." During hemolysis, these young RBCs released a large amount of arginase into the plasma 50. This plasma arginase consumes plasma L-arginine, the substrate required for NO production by endothelial cells, and, in conjunction with the consumption of endothelial NO by cell-free plasma Hb, reduces NO bioavailability 51, 52."

Sildenafil and Tadalafil do not increase NO bioavailability, they inhibit PDE5, which prevents the degradation of cGMP, the end-target signaling molecule that is responsible for many of the cellular and vascular mechanisms mediated by NO.

Author’s Response: We have corrected this in the text.

Ref 63 page 4 lines 176 -178. If hemopexin can disrupt the cf-Hb interactions with HMGB1 to reduce macrophage activation, then hemopexin has anti-inflammatory properties that go far beyond binding and removing heme.

Author’s Response: We agreed with the reviewer that hemopexin may have yet-to-be identified anti-inflammatory properties in addition to binding heme. However, the paper referenced here (now reference 65 in the revised manuscript, PMID: 34478834) did not study HMGB1, measure its levels, or examine the interaction between HMGB1 and cell-free Hb interactions. Given the paper only investigated the effect of hemopexin therapy on the development of pulmonary hypertension and pulmonary fibrosis in a sickle cell disease mouse model, we are unable to comment on the role of HMGB1 and macrophages in SCD or on the anti-inflammatory properties of hemopexin.

Page 5, line 183. Recent advances in understanding (in what?) – what did you have to understand to know that inflammation and thrombosis are correlated? I assume they correlated to each other, but they could also be correlated to something else in how the sentence was written.

Author’s Response: We thank the reviewer for this feedback. We have rewritten this sentence, and it now reads as follows: " Accumulating evidence from human studies discussed below suggests that inflammation is a risk factor for thrombosis."

I did not check all references to determine if they were quoted accurately. However the misstatements associated with the Claudia Morris’ manuscript suggest that the authors fundamentally misunderstood her work. That being the I would ask that the text in the manuscript be checked for accuracy for all references quoted.

Author’s Response: We have checked all references cited in the manuscript for accuracy and confirmed.

Round 2

Reviewer 2 Report

The authors addressed all of my concerns. I must apologize to them for misquoting a reference.  Here are the two sentences.

"However, one of the first papers examining hemopexin in sickle cell disease discovered that cf-Hb interacts with HMGB1 to form an inflammatory complex that binds to TLR4, which induces synergistic increases in cytokine generation by macrophages (3). Importantly, cell culture studies showed that the interaction between cf-Hb and HMGB1 could be disrupted by hemopexin (3)."

Reference 3 should be Lin, et al. as below.  

Lin T, Sammy F, Yang H, Thundivalappil S, Hellman J, Tracey KJ, Warren HS. Identification of hemopexin as an anti-inflammatory factor that inhibits synergy of hemoglobin with HMGB1 in sterile and infectious inflammation. J Immunol. 2012;189(4):2017-22. PMID: 22772444. 

You might consider adding this original reference reporting that cf-Hb forms a complex with HMGB1 in SCD. It could be very important in the future for treating SCD since clinical studies using hemopexin (CSL889) are ongoing.  I have no stake in the company that makes CSL889.  

Author Response

Reviewer 2

The authors addressed all of my concerns. I must apologize to them for misquoting a reference.  Here are the two sentences.

"However, one of the first papers examining hemopexin in sickle cell disease discovered that cf-Hb interacts with HMGB1 to form an inflammatory complex that binds to TLR4, which induces synergistic increases in cytokine generation by macrophages (3). Importantly, cell culture studies showed that the interaction between cf-Hb and HMGB1 could be disrupted by hemopexin (3)."

Reference 3 should be Lin, et al. as below. 

Lin T, Sammy F, Yang H, Thundivalappil S, Hellman J, Tracey KJ, Warren HS. Identification of hemopexin as an anti-inflammatory factor that inhibits synergy of hemoglobin with HMGB1 in sterile and infectious inflammation. J Immunol. 2012;189(4):2017-22. PMID: 22772444.

You might consider adding this original reference reporting that cf-Hb forms a complex with HMGB1 in SCD. It could be very important in the future for treating SCD since clinical studies using hemopexin (CSL889) are ongoing.  I have no stake in the company that makes CSL889. 

Author’s Response: We thank the reviewer for providing the correct reference. This paper (PMID: 22772444) was not a study on the role of HMGB1 in cardiopulmonary complications of sickle cell disease or in macrophages isolated from a sickle cell mouse model. The paper was on the interaction between cell-free hemoglobin and HMGB1 in proinflammatory cytokine production by macrophages isolated from wild-type mice, and it only mentions sickle cell crisis once in the discussion section. Though the focus of this manuscript is specifically on the role of inflammation in cardiopulmonary complications of SCD, we have included this paper in the manuscript in order to move this process forward. We have included a statement on page 5 from lines 204–211, which reads as follows: "Another study in murine macrophages has shown that cell-free hemoglobin and free heme act in synergy with HMGB1 to activate proinflammatory cytokine production in wild-type murine macrophages, and treatment with hemopexin abolishes this interaction 81. Furthermore, treatment with hemopexin significantly suppressed the synergistic production of proinflammatory cytokines, suggesting an anti-inflammatory property of hemopexin 81. This anti-inflammatory ability of hemopexin, in addition to its heme-scavenging function, may provide another potential therapeutic option for addressing inflammation in SCD.”